# Primary care providers' and nurses' knowledge, attitudes, and skills regarding latent TB infection testing and treatment: A qualitative study from Rhode Island

Daria Szkwarko[1]*, Steven Kim[2], E. Jane Carter[3], Roberta E. Goldman[1]

1 Department of Family Medicine, The Warren Alpert Medical School of Brown University, Providence, RI, United States of America, 2 The Warren Alpert Medical School of Brown University, Providence, RI, United States of America, 3 Department of Medicine, The Warren Alpert Medical School of Brown University, Providence, RI, United States of America

☯ These authors contributed equally to this work.

* Szkwarkd@gmail.com

## Abstract

### Background

Untreated latent tuberculosis infection (LTBI) is a major source of active tuberculosis disease in the United States. In 2016, the United States Preventive Services Task Force (USPSTF) recommended that screening for latent tuberculosis infection among individuals at increased risk be performed as routine preventive care. Traditionally, LTBI management–including both testing and treatment–has been conducted by specialists in the United States. It is believed that knowledge gaps among primary care team members and discomfort with LTBI treatment are significant barriers to LTBI management being conducted in primary care.

### Methods and objectives

This qualitative study sought to evaluate the knowledge, attitudes, and skills of primary care team members regarding the LTBI care cascade, and to identify each stepwise barrier limiting primary care teams in following the USPSTF recommendations.

### Results

We conducted 24 key informant interviews with primary care providers and nurses in Rhode Island. Our results demonstrate that overall, few primary care providers and nurses felt comfortable with LTBI management, and their confidence and comfort decreased throughout the cascade. Participants felt least confident with LTBI treatment and held misconceptions about LTBI testing, such as high cost. Although participants were not confident about LTBI treatment, most were enthusiastic about treating patients if provided additional training. Participants suggested that their lack of knowledge regarding LTBI treatment led to high rates of referral to specialist providers.

**Data Availability Statement:** Data cannot be shared publicly as per our Kent Hospital Institutional Review Board approved protocol and

approved consent form. Our approved consent specifically states that participant data that are potentially identifiable will not be shared. Given the nature of our qualitative transcripts and coding sheets, although stripped of identifiers such as names, they contain potentially identifiable and sensitive participant information in even the most aggregate form if shared with certain individuals who are familiar with our context. For further questions regarding the IRB approved protocol and consent, you may contact Dorinda Williams, the Kent Hospital IRB coordinator at dorwilliams@carene.org. Given the nature of our protocol and consent as described above and the potential to share sensitive human subject information, data access requests by qualified researchers can be made to Lynn Menatian at JMenatian@carene.org for further consideration.

**Funding:** DS is partially supported by an internal Institutional Development Award Number U54GM115677 funded by the National Institute of General Medical Sciences of the National Institutes of Health, which funds Advance Clinical and Translational Research (Advance-CTR https://www.brown.edu/initiatives/translational-research/home). Advance-CTR funded this research and members of their team provided mentorship to authors regarding Nvivo use. SK is supported by an internal program that is funded by the NIH/NIAID under R25AI140490 – Emerging Infectious Disease Scholars (EIDS https://www.brown.edu/academics/medical/plme/current-students/enrichment-activities/research-opportunities/emerging-infectious-diseases-scholars-) Program at Brown University. EIDs provided funding to support SK's time, but otherwise did not support the research in any other way. The content is solely the responsibility of the authors and does not necessarily represent the official views of the National Institutes of Health, Advance-CTR, or EIDS.

**Competing interests:** The authors have declared that no competing interests exist.

## Conclusion

The gaps revealed in this study can inform training curricula for primary care team members in Rhode Island and nationally to shift the USPSTF policy into practice, and, ultimately, contribute to TB elimination in the United States.

## Introduction

It is estimated that 13 million people in the United States (US) have latent tuberculosis infection (LTBI). Untreated LTBI is responsible for 80% of tuberculosis (TB) disease nationally [1]. Since the 1960s, preventive therapy (i.e. isoniazid) has been recognized to significantly reduce progression to TB disease. However, only three quarters of at risk individuals are screened for LTBI, and only 62% of individuals who started LTBI therapy completed treatment [2].

In 2016, the United States Preventive Task Force (USPTF) provided updated recommendations that LTBI testing be performed in populations at increased risk as part of routine preventive care [3]. Traditionally, LTBI testing and treatment in the US has often been conducted by specialists in TB clinics. The most common risk factor in the US for TB is having been non-US born; this population already faces many barriers to accessing health care [4]. As stated by Katrak and Flood in a 2018 commentary, it is believed that knowledge gaps among primary care providers (PCPs) and lack of familiarity with LTBI treatment regimens (recommended regimens in the US include 9 months of isoniazid, 4 months of rifampin, 3 months of weekly rifapentine and isoniazid) remain significant barriers to treatment for underserved populations [5]. Furthermore, non-US born populations such as refugees and asylum seekers can experience difficulty in connecting with other services, and it has been suggested that this can be worsened both by poor communication among providers as well as complex healthcare systems [6].

Task shifting LTBI testing and treatment into primary care represents a critical strategy towards TB elimination [7]; patients can be screened and treated within their primary care home, avoid the need to establish care with new specialist providers, and complete documentation of this intervention within their primary care medical record. This can all be done as part of their routine primary care preventive visits. However, prior to the USTPF guidelines update there was little impetus to include LTBI testing and treatment in routine preventive screening and primary care visits. The shift of the USPSTF screening guidelines to a Grade B recommendation ensures private insurance coverage for these activities, thus removing the cost barrier [8]. In order to design educational programs to democratize LTBI knowledge and assist primary care practices and healthcare systems to support primary care providers in LTBI testing and treatment, more information is needed to understand how PCPs and nurses perceive their knowledge, attitudes, and skills regarding LTBI testing and treatment in their practices. Given the exploratory nature of this topic, qualitative methodology can best elucidate the thematic knowledge, attitudinal, and skill gaps that exist.

In this qualitative study, we aimed to explore the knowledge, attitudes, and skills (KAS) among Rhode Island (RI) PCPs and nurses regarding the latent TB infection care cascade—from recognizing and screening the at-risk population for LTBI, to evaluating for active TB disease, and to initiating, monitoring, documenting, and reporting completion of therapy. At each cascade step, there are potential KAS gaps that can influence a PCP's or nurse's ability to carry out LTBI testing and treatment. We set out to use qualitative research methods which are uniquely suited to exploratory research aimed at uncovering these gaps. We then aimed to

apply our findings to facilitate transition of USPSTF guidelines into practice, with the ultimate goal to improve TB prevention and care.

## Methods

This qualitative study was the first phase of a three-phase exploratory sequential translational study (clinicaltrials.gov ID: NCT04188041). A purposive sample of key informant PCPs and nurses working in RI were recruited to participate [9]. Key informant interviews were conducted in order to purposively include PCPs and nurses who work and have an understanding of at-risk populations. Our study reporting conforms to the Consolidated Criteria for Reporting Qualitative Research (S1 Checklist).

### Setting

Key informant interviews (KIIs) were conducted with PCPs and nurses who work in a variety of outpatient practices throughout RI. In the United States, primary care is provided by PCPs which include both physicians as well as mid-level providers such as physician assistants or nurse practitioners. For LTBI care, PCPs can order LTBI testing, order and interpret LTBI evaluation, initiate LTBI treatment, and conduct follow up visits. Nurses work alongside PCPs on the primary care team where their role involves ordering LTBI tests, including planting and interpretation of tuberculin skin tests, and conduct treatment adherence assessment during follow up visits. The majority of interview participants work in primary care settings that serve large non-US born populations. RI has one TB specialty clinic, located in Providence, the largest city in the state, that provides treatment for both TB disease and latent TB infection [10]. In 2019, there were 14 individuals diagnosed with TB disease, and 93% of these individuals were non-US born [11]. Although LTBI is a reportable condition in RI, compliance with this reporting requirement is unclear. Traditionally, many primary care practices in RI refer people with positive LTBI screen to the TB specialty clinic; however a small proportion of patients are diagnosed and treated solely through community health centers that commonly serve LTBI-affected populations [12].

### Participant recruitment

Using purposive, criteria-based sampling to recruit PCPs and nurses, we sent emails to invite them to participate in the study [9]. Our inclusion criteria were mid-level PCPs such as nurse practitioners as well as physicians with family medicine or internal medicine training. We excluded those who did not currently have a primary care panel. We used purposive sampling to ensure inclusion of participants from Federally Qualified Health Centers in the state that serve at-risk populations, as well as other practice setting types. Authors (DS and REG) are faculty in the department of Family Medicine, and many PCPs throughout the state are alumni from this department. In addition to emailing our alumni and departmental networks, we circulated study information to medical directors at Federally Qualified Health Centers (FQHCs) and other large practices in the state that serve non-US born populations to recruit PCPs and nurses. Participants were offered a $75 gift card for their participation.

### Instrument and data collection

A qualitative, semi-structured interview guide was developed based on a literature review and authors' clinical expertise regarding potential knowledge, attitude, and skill gaps. The guide included open-ended core questions that were supplemented with spontaneous probes and follow-up questions during interviews to explore the barriers impacting all LTBI steps (S1

File). We piloted the interview guide with two PCPs and modified the questions based both on their feedback and the team's input. After obtaining consent from participants, KIIs were conducted either in person or virtually by authors SK or DS. SK is an Asian-American, cis-male fourth-year medical student and DS is a white, cis-female family medicine and preventive medicine physician. Both interviewers underwent qualitative interviewer training with REG, a qualitative research expert with more than 20 years of experience. After the training, REG also listened to the pilot interviews and provided feedback on interviewer technique. For in person interviews, written consent was obtained in person. For virtual interviews, participants were emailed a consent document in advance to review. The consent document was read aloud at the start of the virtual interview, participants were invited to ask any clarifying questions, and verbal consent was documented by the study team by signing the informed consent on the participant's behalf and indicating that consent was obtained verbally. We continued data collection until we reached data saturation. We identified when data saturation was reached through iterative discussion of the interviews by the two interviewers throughout the interview process. During this process, we identified when the same responses were repeatedly elicited and no new content was being collected, or themes emerging [13].

## Data analysis

Interviews were digitally audio-recorded and then professionally transcribed verbatim. Transcripts were cleaned by SK; any identifiable data were removed, any inaudible text was identified, audio-recordings were listened to, and transcripts were edited to ensure accuracy. Two data analysis approaches were used in stepwise fashion. First, Immersion/Crystallization was used as transcripts were read in their entirety by both DS and SK, while taking notes for overall familiarity with content [14]. These notes were then used as a basis for Template Organizing Style Analysis [15]. whereby a codebook with code definitions was created based on the knowledge, attitude and skill constructs at each cascade step that informed the interview guide, as well as content categories and themes that emerged in the first stage of Immersion/Crystallization analysis of the transcripts [14]. The codebook was tested independently by SK and DS on three transcripts. Coding was compared and discussed, and the codebook was finalized. Flexible coding schemes were maintained to accommodate addition of new codes during the coding process. DS completed the coding process using the data analysis software, NVivo [16]. Following coding, DS and SK again used Immersion/Crystallization techniques to read each code report for content, patterns and themes [14]. This process, along with discussing emerging findings throughout the process with all co-authors, resulted in final interpretation of the data. Our stepwise data analysis approach, the use of NVivo, and our check-ins with all co-authors ensured rigor at each stage of analysis.

## Ethics approval

This study was approved by the Kent Hospital Institutional Review Board in RI. Lifespan and Brown University entered into an IRB reliance agreement with Kent Hospital with both organizations relying on Kent Hospital, as the IRB of record, for IRB review and oversight of the project.

## Results

We conducted 25 interviews—22 PCPs and 3 nurses. One physician interview was excluded, as the interviewee had not yet started working in a primary care setting. Of the 21 PCPs included, 18 (86%) were trained in family medicine, two (9%) in medicine/pediatrics, and one (5%) in internal medicine. Seven (33%) practiced in FQHCs, five (24%) in academic practices,

and nine (43%) in private practices. Three of the PCPs were resident physicians in training. The three registered nurses worked at an FQHC with a clinic that conducted health screening for immigrants to the US.

Most PCPs practiced full spectrum primary care, with 19 (90%) seeing patients of all ages and 20 (95%) seeing pregnant patients. Of all PCPs and nurses, 11 (46%) reported many of their patients are Spanish speaking, 19 (79%) reported that more than one third of their panel is non-US born, and 10 (42%) reported they see many patients with lower socioeconomic status (Table 1).

## Overall LTBI management

Overall, few participants (PCPs and nurses) felt comfortable with LTBI management in general, and confidence and comfort decreased as the care cascade progressed (i.e. participants felt most comfortable and confident with testing and least comfortable and confident with treatment), Fig 1.

Those with unique personal experiences with their own LTBI diagnoses, work experiences in high TB and LTBI burden settings, and/or training or mentorship regarding LTBI expressed more knowledge and comfort with LTBI management. Residents in training expressed more comfort with looking up LTBI information on resources such as UpToDate [17] than did experienced PCPs. One PCP stated:

> "I grew up in a TB endemic country. So when I was doing medical training we did have a lot of TB patients in the hospital where I was trained. And then when I did my residency in New Hampshire we had a relatively large refugee population. So many of them had latent TB, or even some of them had active TB. . .there we didn't have a TB clinic so we treated them directly."

**Table 1. Demographics of primary care providers and nurses interviewed (n = 24).**

| Characteristics | Providers (n = 21) | Nurses (n = 3) |
|---|---|---|
| Gender Female | 11 (52%) | 2 (66%) |
| Year of residency training completion (median and IQR) | 2017 [2014, 2018] (missing = 3 who were in residency training) | N/A |
| Family Medicine trained | 18 (86%) | N/A |
| Resident Physicians | 3 (14%) | N/A |
| FQHC | 7 (33%) | 3 (100%) |
| Academic Practice | 5 (24%) | 0 (0%) |
| Private Practice | 9 (43%) | 0 (0%) |
| Participant cares for patients of all ages | 19 (90%) | 3 (100%) |
| Provider care for pregnant patients | 20 (95%) | 3 (100%) |
| Large proportion (>1/3) of patient panel is Spanish speaking | 9 (43%) | 2 (100%) (missing = 1) |
| Large proportion (>1/3) of patient panel is non-US born | 17 (81%) | 2 (66%) |
| Large proportion of patient panel with lower SES | 10 (48%) | 2 (66%) |

FQHC = Federally Qualified Health Center; IQR = interquartile range; SES = socioeconomic status.

**Fig 1. Primary care providers' and nurses' knowledge, attitudes, and skills regarding LTBI management–major themes throughout the care cascade.**

Regarding USPSTF guidelines, 9 out of 16 participants who engaged in discussion about the guidelines were unaware of the current recommendation for LTBI screening and treatment.

Most PCPs expressed discomfort with evaluating and treating LTBI in special populations such as patients living with HIV, pregnancy, children, and the elderly. For example, one PCP stated:

> "I think– the ones that are harder [to evaluate and treat for LTBI] are the older patients who have a lot of comorbidities. So I am not comfortable with HIV in general because I don't see very much of it. And I'm not conversant in [HIV] drugs, and I know they have a lot of drug interactions. So I certainly would feel more uncomfortable with the evaluation than diagnosing them with latent TB as much as I would feel really uncomfortable with the decision-making around treatment."

Another PCP mentioned, "I think if [patients] had underlying immunodeficiency, and probably the pediatric population, I would feel less comfortable diagnosing active TB or ruling out active TB." Resident physician participants were more willing to attempt all aspects of LTBI care given that they felt supported in a training environment. One said, "I'm still a resident, meaning I'm in a learning environment. So I have multiple colleagues that I can ask questions to, but I think once you're out of residency it's a little harder."

We report the rest of our findings according to the LTBI care cascade steps.

## LTBI screening

The majority of PCPs screened for LTBI very infrequently, with three reporting that they never screen for LTBI. The few who claimed to be screening regularly work in unique practices that see newly arrived immigrants more frequently when compared to other practices. One PCP described seeing many new immigrants from Central America in her patient panel, stating "I've only been working for like two months. I probably see fifteen to twenty patients a week right now. Out of those patients that come—maybe five out of fifteen are screened."

Most participants knew several factors that should prompt LTBI screening. However, few were aware of all of the factors. The majority mentioned immigration from or travel to a high TB burden country to be a reason to conduct LTBI testing. For example, one PCP mentioned:

"I generally think [that LTBI screening factors are] something I would probably at this point need to look up, but thinking of people who have come from or have family members, you know, first generation or visit countries with high prevalence of TB, which I think of generally as certain countries in Central or South America, Eastern Europe, Asia...I would again kind of look up what are the top countries that are recommended [for screening]."

Many PCPs identified incarceration, employment screening, exposure to active TB disease, and being a healthcare worker as reasons to screen for LTBI. Few mentioned immunocompromised conditions like HIV or starting an immunosuppressant as an indication to screen. PCPs demonstrated gaps in their knowledge regarding LTBI screening, as some mentioned symptoms of active TB disease as an indication to screen, and others were unsure about guidelines regarding screening healthcare workers.

When PCPs proceeded to test for LTBI, the majority preferred the Interferon Gamma Release Assay (IGRA) over purified protein derivative (PPD). They perceived IGRA to be more accurate and easier to perform and interpret. Most acknowledged the possibility of a false-positive reaction with the PPD for patients with history of Bacillus Calmette-Guérin (BCG) vaccination. One PCP mentioned, "But if a patient has a BCG vaccine growing up in a country where that's done, routinely the PPD is likely to have a false positive. So then the IGRA is a more accurate test. And generally I just find it somewhat more convenient [for the patient] than having two visits." The most commonly reported benefit of the IGRA was its superior convenience for patients and team members, as it could be performed over a single visit, as opposed to two visits with the PPD. A PCP said, "I think the IGRA is great because they order labs on people all the time for less important things. I can add it on with other labs, or it's very easy to do on a Thursday at 3:00 p.m. when you don't want to have them come back to schedule a PPD and another visit." A nurse commented, "We had a bunch of times where [patients] come back a week later. They're like, 'Oh yeah, I'm here for my [PPD read].' It's like, 'We can't read it now—you have to have it replanted.' So we have to see them a week from their plant date and replant it."

Furthermore, PCPs often felt undertrained on PPD planting and interpretation. One PCP stated:

"I also know that [when] PPD testing is read in different patient populations that the reaction—the measurement of the millimeters—is interpreted differently. So the cutoff for how much reaction you get in HIV is very different than the cutoff in an asymptomatic, well patient. And so it's that variability to it that I don't like."

Another provider mentioned:

"Yeah. I find interpreting the induration challenging. When you're interpreting, marking down where the induration begins and where it ends on the patient's arm is –I think is a little subjective...I suspect that there's probably a lot of inter-reader variability that may affect the results of the PPD testing."

Participants who were more comfortable with the PPD had unique patient populations that necessitated more on-the-job exposure to PPDs than most of the other participants

interviewed had. Additionally, nurses comprised the majority of participants who were comfortable with planting and interpreting the PPD. One nurse shared, "I feel like I was pretty comfortable, you know, just having worked at the hospital. . .the specific experience of immigration clinic, I definitely gained a lot more confidence [in planting and interpreting PPDs] as a result of that experience."

Despite the many benefits discussed about IGRA testing, PCPs demonstrated gaps in knowledge regarding LTBI testing in general. For example, PCPs were largely unaware that the effect of BCG on PPD wanes over time or that there are different age recommendations for LTBI testing (e.g. IGRA is not recommended under age two). One PCP said, "I think I'm recalling that if someone has had the BCG vaccine, that IGRA is better or more accurate. And I forget about the ages. I think there are age recommendations when you do one versus the other like in kids, for example." Moreover, some PCPs expressed the misconception that patients with a positive LTBI test result can test negative later in life.

PCPs were often concerned about the cost of the IGRA test compared to the PPD. Cost alone was sometimes the primary driver towards planting a PPD vs ordering an IGRA. One PCP stated, "Even though there's the difficulties of returning, I try to start with the PPD mostly just because of the cost of the IGRA versus the PPD." A nurse stated:

> "Yes, so sometimes it's an uninsured [patient] and they didn't want to pay the cost of it. Especially when people are trying to get a job, and they don't have insurance yet with their job. And then the job is telling them they need to have PPD testing or TB testing. So then we're ending up– a lot of times doing a PPD for that situation because it's much cheaper for them."

PCPs occasionally discussed treating the IGRA as a confirmatory or second line test to the PPD. One PCP stated, "So for patients who have been vaccinated with BCG's or who previously tested positive for PPD. . .We would normally order a Quant Gold [IGRA] anyways and have that confirmatory backup."

Many PCPs mentioned that they report positive LTBI results to the Rhode Island Department of Health (RIDOH). Those who report consistently have nursing support to complete the reporting process. Of those who did not report LTBI to the RIDOH, most were not aware of reporting guidelines. A few assumed that the TB clinic reports to the RIDOH and referred to them without reporting.

## TB disease evaluation

Many participants were comfortable with conducting TB disease evaluation (symptom screening and ordering a chest x-ray to rule out active TB disease) for otherwise healthy adult patients who have a positive LTBI screening test. However, the majority refer otherwise healthy adult patients to the TB clinic for final diagnosis interpretation and management. One PCP said, "I think probably the most common thing I now see is positive testing, negative chest x-ray, send them to the TB Clinic to do all other treatment and management decisions." Some participants, particularly nurses, expressed comfort with TB disease evaluation because they use a symptom checklist either from their electronic medical record or the CDC website to conduct TB disease evaluation. One described the process as, "We have a set form in our computer system where we literally just go down a list. So I felt pretty comfortable."

A few PCPs avoided evaluation altogether and instead immediately referred to the TB clinic. One PCP stated, "Just because I don't [do the TB evaluation] ever, I would probably send to the [TB Clinic]. If for some reason I couldn't get them there, I don't know, [they are]

homebound or something, I would probably go to UpToDate." A few participants mentioned language as a barrier to evaluation. The participants who expressed discomfort with LTBI evaluation also noted the lack of patients with LTBI in their practices. "I wouldn't say super comfortable, especially after four years of not having to deal with positive test results."

## LTBI treatment

Generally, participants were not knowledgeable about or comfortable with providing LTBI treatment. While many PCPs were aware of some LTBI medications, including isoniazid and rifampin, they were less familiar with the variety of side effects and medication interactions associated with these medications. PCPs were also unsure about treatment duration and guidelines. Few knew the appropriate timing of follow up and appropriate labs or symptoms to monitor for. Very few knew about combined shorter course regimens (i.e. isoniazid and rifapentine for three months) as an available regimen. When asked about why PCPs are not treating LTBI in primary care settings, one PCP stated:

> "So one might just be lack of knowledge that they can prescribe those medications. Two might be fear of, like, do I know the treatment regimen, do I not know the treatment regimen? Even if I know it, I think there are a lot of potential side effects. I mean the risk is low, but the severity of the side effects with isoniazid is, and plus or minus rifampin, can be serious. So I think that sometimes people might have a heightened fear. It's more fear-based rather than, like the actual risk is this low, and they might not know side effects and the monitoring."

Many providers also expressed discomfort with treating patients with LTBI. One stated, "I think once you're out of residency it's a little harder. It's your license on the line if you mess it up, and if you haven't previously done it [treated LTBI], then it feels like more of a risk to try. And you know, it's a long, extensive therapy." Some struggled to confidently decide which patients are appropriate candidates for treatment. Other PCPs were uncomfortable treating LTBI in patients living with other medical considerations such as HIV, pregnancy, or chronic illness. Moreover, PCPs felt under-supported and under-trained to shoulder the added risk and time associated with LTBI treatment. A few PCPs mentioned that treatment would go against clinic policies or established tendencies to refer to specialists.

> "I think it is just kind of the policy—how people work in my clinic. They refer the patient to the TB clinic. Myself, I think when I started the practice here I did tell my team, 'Well I would like to be treating them, so if possible I can treat them myself.' But they just always [said]—we send them to the TB Clinic. . .Personally I would like to treat them myself."

In addition to feeling less comfortable with LTBI treatment, PCPs also felt that specialists and the TB clinic had more resources to provide better patient care. One PCP stated:

> "I think it's also a time issue. I think when you're also being asked to screen for cancer and screen for other chronic conditions and also the patient has depression, and also they don't have money to pay for their drugs. I think the sheer volume of what we're addressing in primary care and the amount of time we're being asked to do it in is so incongruous with each other that I think when you have a resource that's set up and specialized for this, something like a TB clinic, it's very easy to defer to treatment elsewhere."

PCPs expressed that specialists had more time, skill, and knowledge to appropriately educate and follow up with LTBI patients. One stated:

"I mean if you're operating in a fee for service world which we're still in. I know it's transitioning out. But you know, to spend five minutes with a patient and say I'm going to refer you to the TB clinic is easier than spending fifteen, twenty minutes with a patient and counseling them about all the different pros and cons and follow-up plans and reasons for this and that. I just think that from a financial standpoint it's more efficient to just refer somebody out and go to your next patient than to spend the time trying to explain to the patient what it means and what we need to do next to treatment options."

Some PCPs mentioned fear of conflict with specialists. One stated, "I think TB treatment has sort of been owned by global health and infectious disease for such a long time that I think you're stepping on toes. You're creating turf battles if you do it." However, some PCPs expressed concern that patients who are referred are often unable to readily access specialist care. One commented, "But it's a lot of work to ask [patients] to go to another doctor, and there's transportation issues. There's scheduling issues. And our patients don't have stable phone numbers and don't check their voicemail the way that I think a lot of us, if it were you or I who were going through this process."

Many participants expressed interest in being able to provide LTBI treatment to patients. One mentioned, "I mean if I could, then I would like to treat them myself because I know the patient better, and I can make sure that they are actually taking the medication." Another said, "I'd like to learn more. I know that there's an option for a shortened course. I believe it's isoniazid and rifampin together. I don't prescribe rifampin. I don't have a lot of experience or comfort with it. I think it can turn your eyes or tears orange or something like that."

Regarding LTBI treatment follow up, many participants knew that they should be looking out for medication side effects and conducting follow-up visits with patients. However, few actually knew specific side effects or time intervals for follow-up. Many were not aware of the current guidelines regarding laboratory testing prior to and during LTBI treatment. One stated, "Yeah I mean I would have to look this up, but I think you have monthly visits to make sure there are no issues taking the medication. I forget what the recommendation is on checking liver enzymes. Maybe that would be something I would be doing."

## Discussion

In order to successfully adopt the USPSTF's updated guidelines recommending LTBI screening as part of routine preventive screening in primary care, it is critical to understand PCPs' and nurses' knowledge, attitudes, and skills regarding LTBI management. The current LTBI management model depends heavily on referrals to specialists, is complicated by significant loss of patients throughout the process, and creates both time and cost burdens for patients. Benefits in recognizing the central role played by primary care teams in streamlining patient-centered care are clear, but the inclusion of primary care for LTBI screening and treatment represents a paradigm shift for public health departments, specialists, and primary care teams. As a whole, our study demonstrates that PCPs are uncomfortable with LTBI care in general, and their confidence and comfort decreases throughout each step of the LTBI management cascade—screening, evaluation for TB disease, and treatment [2]. This manuscript is the first qualitative study to our knowledge to explore this topic and sheds light on the limitations surrounding adoption of the USPSTF LTBI guidelines among primary care team members.

In regard to LTBI screening, PCPs were familiar with both testing modalities—the PPD and IGRA—but many held common misconceptions about them. The most common misconception regarding the IGRA was that it is costlier to perform than the PPD. Although out-of-pocket charges for IGRA over PPD does represent a significant difference, this cost differential pertains to uninsured patients, as screening test recommendations under USPSTF guidelines are covered by insurance providers. In addition, the IGRA has actually been proven to be cost-effective for practices seeking to save costs related to screening and treating at-risk individuals such as non-US born populations [18].

The most common misconception that PCPs held regarding the PPD was that a childhood BCG vaccination can interfere with a PPD result throughout a patient's lifetime. Though the BCG is associated with a positive PPD result in childhood, the effect wanes with time and is likely negligible at ages over 30 years old [19]. Another misconception was that a positive LTBI test result can convert to negative later in life. Misconceptions such as this one can lead to unnecessary testing and potential harm since repeating a positive PPD for example can lead to a larger, more painful, reaction. Generally, PCPs believed that though the IGRA was the more accurate test, it should be treated as a confirmatory or second line test to the PPD. On the contrary, the Centers for Disease Control recommends IGRA screening over PPD in non-US born individuals from countries with moderate to very high incidence of TB disease who have received BCG vaccination. It does not recommend utilizing both tests when testing for LTBI [20].

PCPs and nurses said they were often comfortable with evaluating young and healthy patients for TB disease. However, following a positive test result, PCPs were then hesitant to proceed to treatment, often referring at this point to specialists for further management. While most PCPs did not have the knowledge or comfort to treat LTBI in primary care, the majority expressed enthusiasm to learn and incorporate treatment into their practices, especially for otherwise healthy adults. Similar to the transition of Hepatitis C management to a primary care setting, a shift in LTBI management to primary care could allow for reduced loss to follow-up and allow specialists to focus on patients with increased risk or complicating co-morbidities [21]. Additionally, PCPs often have strong physician-patient relationships [22], and relationship strength has been shown to increase medication compliance, which may be particularly important for the vulnerable populations that LTBI disproportionately affects [23]. Diagnosing and treating LTBI prior to its progression to TB disease allows for substantial cost savings, as LTBI carries 5–10% lifetime risk of progression to TB disease, but its treatment is at least 32 times less expensive than the treatment of TB disease [24]. It is also well understood that it will be impossible to reach TB elimination without improvement in LTBI treatment completion [25].

This study has several limitations. Given that our participants were all PCPs or nurses in Rhode Island, their views may differ from PCPs and nurses in other states nationally. Most of the PCPs were family medicine trained and connected to the Warren Alpert Medical School of Brown University network. They may reflect different values when compared to PCPs trained in other specialties or in non-academic settings. Three PCPs were trained in internal medicine and two in pediatrics, and although their views were consistent with FM providers' perspectives, future research should include a wider variety of primary care specialties including OBGYN. Moreover, most participants completed training recently. Although this suggests that LTBI training in medical school, residency and nursing school is lacking, their perspectives may not be reflective of PCPs and nurses who are further out of training. We were unable to recruit nurses beyond one health center and this health center conducted immigration screening. Therefore, nurses in our study seemed well-versed in LTBI testing which may not be consistent with nurses in clinics serving fewer non-US born patients.

The major strength of this study is that it reveals specific knowledge gaps among PCPs and nurses that could be targeted in future training programs. Our study suggests that the preconceived belief that PCPs are unfamiliar with LTBI treatment is likely true. Overcoming this barrier with increased training as well as improved primary care infrastructure is key to TB elimination in our country.

## Conclusion

PCPs and nurses in Rhode Island have limited confidence and comfort regarding LTBI management and this decreases throughout the care cascade. However, participants expressed readiness to learn how to integrate these aspects of care into their practices. The lack of knowledge regarding LTBI treatment leads to high rates of referral to specialty clinics. Non-US born populations are disproportionately impacted by barriers that decrease their ability to see a specialist, thereby increasing loss to follow-up. For the patient, it is a primary prevention intervention lost, and for the public health system, an opportunity missed to contribute toward TB elimination. Increased training focused on the gaps identified in this study has potential to move the USPSTF recommendations on LTBI screening into broad primary care adoption.

## Supporting information

**S1 Checklist. COREQ checklist.**
(DOCX)

**S1 File. Qualitative interview guide for PCPs.**
(DOCX)

## Acknowledgments

We would like to acknowledge all of the busy primary care providers and nurses who took the time to participate in this study. We would also like to thank Thomas Bertrand, Jill Lamantia, and Caroline Gummo from the Rhode Island Department of Health as well as Dr. Natasha Rybak from the RISE TB Clinic for their historical perspective regarding LTBI practices in Rhode Island.

## Author Contributions

**Conceptualization:** Daria Szkwarko, Steven Kim, E. Jane Carter, Roberta E. Goldman.

**Data curation:** Daria Szkwarko, Steven Kim, Roberta E. Goldman.

**Formal analysis:** Daria Szkwarko, Steven Kim, Roberta E. Goldman.

**Funding acquisition:** Daria Szkwarko, E. Jane Carter, Roberta E. Goldman.

**Methodology:** Daria Szkwarko, Steven Kim, E. Jane Carter, Roberta E. Goldman.

**Project administration:** Daria Szkwarko, Steven Kim.

**Supervision:** Daria Szkwarko, Steven Kim, E. Jane Carter, Roberta E. Goldman.

**Validation:** Steven Kim.

**Writing – original draft:** Daria Szkwarko, Steven Kim.

**Writing – review & editing:** Daria Szkwarko, Steven Kim, E. Jane Carter, Roberta E. Goldman.

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
