## [Decision Letter · Decision Letter 0]

10 Aug 2021

PONE-D-21-05814

Primary care providers’ and nurses’ knowledge, attitudes, and skills regarding latent TB infection testing and treatment: A qualitative study from Rhode Island

PLOS ONE

Dear Dr. Szkwarko,

Thank you for submitting your manuscript to PLOS ONE. After careful consideration, we feel that it has merit but does not fully meet PLOS ONE’s publication criteria as it currently stands. Therefore, we invite you to submit a revised version of the manuscript that addresses the points raised during the review process.

The reviewers have identified a range of concerns regarding the content and presentation of your manuscript that need to be carefully addressed in a revision. Please pay particular attention to responding to each of the points raised by Reviewer 3.

We look forward to receiving your revised manuscript.

Kind regards,

Jamie Males

Staff Editor

PLOS ONE

Journal Requirements:

3. Please consider including more information on the number of interviewers, their training and characteristics; and please provide the interview guide used.

"DS is partially supported by Institutional Development Award Number U54GM115677 from the National Institute of General Medical Sciences of the National Institutes of Health, which funds Advance Clinical and Translational Research (Advance-CTR https://www.brown.edu/initiatives/translational-research/home). SK is supported by NIH/NIAID under R25AI140490 – Emerging Infectious Disease Scholars (EIDS https://www.brown.edu/academics/medical/plme/current-students/enrichment-activities/research-opportunities/emerging-infectious-diseases-scholars-) Program at Brown University.

The content is solely the responsibility of the authors and does not necessarily represent the official views of the National Institutes of Health, Advance-CTR, or EIDS."

5. Please include captions for your Supporting Information files at the end of your manuscript, and update any in-text citations to match accordingly. Please see our Supporting Information guidelines for more information: http://journals.plos.org/plosone/s/supporting-information

Reviewers' comments:

Reviewer's Responses to Questions

**Comments to the Author**

1. Is the manuscript technically sound, and do the data support the conclusions?

Reviewer #1: Partly

Reviewer #2: Yes

Reviewer #3: Partly

2. Has the statistical analysis been performed appropriately and rigorously? 

Reviewer #1: N/A

Reviewer #2: N/A

Reviewer #3: No

3. Have the authors made all data underlying the findings in their manuscript fully available?

Reviewer #1: Yes

Reviewer #2: Yes

Reviewer #3: Yes

4. Is the manuscript presented in an intelligible fashion and written in standard English?

Reviewer #1: No

Reviewer #2: Yes

Reviewer #3: No

5. Review Comments to the Author

Reviewer #1: Thank you for your valuable work. The following will be effective in improving your work.

1. Explaining the necessity of working with a qualitative method has not been done, while the questions can be answered with a quantitative method.

2. There is a need to edit and justify the lines.

3. What was the criterion for completing sampling and saturation in this study?

4. Which method was used to analyze the data? (With reference)

5. How is the data confidentiality criterion met?

6. Explain the inclusion and exclusion criteria.

7. Explain the questions used in the research (At least the basic questions)

8. Category tagging has happened with quantitative and pre-arranged visibility.

Reviewer #2: This a wonderful study and is aimed at promoting ending of TB disease through not missing any LTBI untreated. It is very vital that this approach to ending TB is widely known and utilised by all countries main with high TB stats.

In the background it was noted that there was no mention of LTBI treatment regimens as to be comprehensive in the clear description of the problem at hand.

There is also a need to describe the distinct difference between the PCPs and Nurses as these causes confusion as to who is who. This is given the notion that nurses are primary health care provided and this is at least what is known universally.

The article was well formatted

Reviewer #3: Manuscript Number: PONE-D-21-05814

Article Type: Research Article

Full Title: Primary care providers’ and nurses’ knowledge, attitudes, and skills regarding latent TB

infection testing and treatment: A qualitative study from Rhode Island

I read carefully the paper with title and manuscript list above.

Beside that authors rise a thematic that is very import for healthcare institutions and hospitals, I consider that the proposal paper lacks in many aspects it orders to meet the scientific requirements for publication, especially in a journal as PLOS One is.

According to my opinion and the main concern is that the number of 21 primary care providers (PCP) including in this quality study is not sufficient, in a population of Rhode Island of more than 1 Mio, and a number of physicians of more than 2500. Also, I consider that the authors, should prepare one questionnaire that is uniform and all participants in the study to answer on it and not to be based just on the interview. I consider that on this approach the authors could get much more reliable data’s for their scientific study and based on these to have much more clear picture for knowledge, attitudes and skills of PCP regarding LTBI in Rhode Island.

In Introduction part there is many sentences that has not a reference; Line 62-65; Line 71-80; Line 84-86; ect…

The Results part has not been clarifying fully in all study aspect and moreover is not present appropriately. There is just one table and some interviews.

The Reference part is not uniform. I’m not sure for the style the authors selected and if this style has been applying to all references. See references Nb. 6,7,8,9,10,11,…

Also, the English written style in some sentences was not clear and understand, so my proposal is to be check by any professional scientific experience researchers for improve.

Based on all point mention above, I considered that this paper should be revise completely and after to be submitted once more.

Thank you and best regards.

6. PLOS authors have the option to publish the peer review history of their article (what does this mean?). If published, this will include your full peer review and any attached files.

Reviewer #1: **Yes: **Ebrahim Aliafsari mamaghani

Reviewer #2: **Yes: **Prof Lufuno Makhado

Reviewer #3: No

---

## [Author Response · Author response to Decision Letter 0]

11 Nov 2021

Please see attached Response to Reviewers document.

---

## [Decision Letter · Decision Letter 1]

21 Jan 2022

PONE-D-21-05814R1Primary care providers’ and nurses’ knowledge, attitudes, and skills regarding latent TB infection testing and treatment: A qualitative study from Rhode IslandPLOS ONE

Dear Dr. Szkwarko,

Thank you for submitting your manuscript to PLOS ONE. After careful consideration, we feel that it has merit but does not fully meet PLOS ONE’s publication criteria as it currently stands. Therefore, we invite you to submit a revised version of the manuscript that addresses the points raised during the review process.

We look forward to receiving your revised manuscript.

Kind regards,

Hamidreza Karimi-Sari, MD

Academic Editor

PLOS ONE

Reviewers' comments:

Reviewer's Responses to Questions

**Comments to the Author**

1. If the authors have adequately addressed your comments raised in a previous round of review and you feel that this manuscript is now acceptable for publication, you may indicate that here to bypass the “Comments to the Author” section, enter your conflict of interest statement in the “Confidential to Editor” section, and submit your "Accept" recommendation.

Reviewer #1: All comments have been addressed

2. Is the manuscript technically sound, and do the data support the conclusions?

Reviewer #1: Partly

3. Has the statistical analysis been performed appropriately and rigorously? 

Reviewer #1: No

4. Have the authors made all data underlying the findings in their manuscript fully available?

Reviewer #1: No

5. Is the manuscript presented in an intelligible fashion and written in standard English?

Reviewer #1: Yes

6. Review Comments to the Author

Reviewer #1: Thank you for your valuable work. The following can be effective in improving your work

The necessity of the work and the reason for using the qualitative method should be highlighted in the introduction.

- Explain the method of analysis and the type of qualitative method.

- Explain how to achieve saturation

- Explain how to ensure study rigor.

- Labeling the categories with a qualitative look

- At the beginning of the method, the clinical trial code is stated, but this study is descriptive and qualitative

---

## [Author Response · Author response to Decision Letter 1]

22 Feb 2022

Response to Reviewers

Reviewer #1: Thank you for your valuable work. The following can be effective in improving your work

The necessity of the work and the reason for using the qualitative method should be highlighted in the introduction.

Thank you for this comment. We have added justification for the work as well as using qualitative methodology in the introduction.

- Explain the method of analysis and the type of qualitative method.

This study used key informant interviews with PCPs and nurses in Rhode Island who work with at risk populations. A justification for this type of method has been added to the method section.

The methods of analysis are described in the data analysis section and involved two steps – immersion/crystallization and template organizing style. We recognized that we accidentally included the immersion/crystallization reference in the wrong place and have updated this.

- Explain how to achieve saturation

The last line of the data collection paragraph describes how we reached data saturation. Reference 13 supports our approach.

- Explain how to ensure study rigor.

Our stepwise data analysis approach, the use of Nvivo, and our check-ins with all co-authors ensured study rigor. We have added a sentence reflecting this at the end of data analysis.

- Labeling the categories with a qualitative look

Our process of using immersion/crystallization for familiarity with content and to develop the code categories for the next phase of template style analysis using NVivo coding is outlined in the analysis section. With this method, we coded for content and then went back into the coded data, again using immersion/crystallization techniques, to identify patterns and themes in the coded data to come to final interpretation.

- At the beginning of the method, the clinical trial code is stated, but this study is descriptive and qualitative

This manuscript describes the results of the first aim of a three-phase exploratory sequential translational study. Aim 2 is a trial. As described in detail when the clinical trial # is searched (https://clinicaltrials.gov/ct2/show/NCT04188041), the results of the qualitative study described in this manuscript informed the design of the educational intervention studied in Aim 2. We did not provide additional details about this since the methods are outlined carefully on the trial website.

---

## [Decision Letter · Decision Letter 2]

28 Feb 2022

PONE-D-21-05814R2Primary care providers’ and nurses’ knowledge, attitudes, and skills regarding latent TB infection testing and treatment: A qualitative study from Rhode IslandPLOS ONE

Dear Dr. Szkwarko,

Thank you for submitting your manuscript to PLOS ONE. After careful consideration, we feel that it has merit but does not fully meet PLOS ONE’s publication criteria as it currently stands. Therefore, we invite you to submit a revised version of the manuscript that addresses the points raised during the review process. Please carefully consider the peer-review comments in the next revision. Please submit your revised manuscript by Apr 14 2022 11:59PM. If you will need more time than this to complete your revisions, please reply to this message or contact the journal office at plosone@plos.org. Please include the following items when submitting your revised manuscript:A rebuttal letter that responds to each point raised by the academic editor and reviewer(s). You should upload this letter as a separate file labeled 'Response to Reviewers'.A marked-up copy of your manuscript that highlights changes made to the original version. You should upload this as a separate file labeled 'Revised Manuscript with Track Changes'.An unmarked version of your revised paper without tracked changes. You should upload this as a separate file labeled 'Manuscript'.If applicable, we recommend that you deposit your laboratory protocols in protocols.io to enhance the reproducibility of your results. Protocols.io assigns your protocol its own identifier (DOI) so that it can be cited independently in the future. For instructions see: https://journals.plos.org/plosone/s/submission-guidelines#loc-laboratory-protocols. Additionally, PLOS ONE offers an option for publishing peer-reviewed Lab Protocol articles, which describe protocols hosted on protocols.io. Read more information on sharing protocols at https://plos.org/protocols?utm_medium=editorial-email&utm_source=authorletters&utm_campaign=protocols.

We look forward to receiving your revised manuscript.

Kind regards,

Hamidreza Karimi-Sari, MD

Academic Editor

PLOS ONE

Journal Requirements:

Reviewers' comments:

Reviewer #1: For The necessity of the work and the reason for using the qualitative method should be highlighted in the introduction:

Your changes were not convincing to me. You can highlight the need to use this approach.

For Explain how to ensure study rigor:

Your changes were not convincing to me. Criteria as well as action for each criterion are not mentioned

For Labeling the categories with a qualitative look:

Your changes were not convincing to me.

Needs editing, especially in the case of spacing.

---

## [Author Response · Author response to Decision Letter 2]

30 Mar 2022

Journal Requirements:

-Thank you for these comments. We have not retracted any references. However, we had added several new references to support our responses in the previous revision. We have indicated relevant references below in our responses to reviewer comments.

Reviewer #1: For The necessity of the work and the reason for using the qualitative method should be highlighted in the introduction:

-Your changes were not convincing to me. You can highlight the need to use this approach.

Thank you for your comment. Regarding the necessity of the work, we would like to highlight included references that establish the importance of LTBI screening in high risk populations (See Reference #4) and the need to task shift LTBI management into primary care (See Reference #7). You can also see the reference for the United States Preventive Services Task Force’s recommendation regarding LTBI screening (See Reference #8 and #3). As mentioned in these references, LTBI screening and treatment is critical to achieve TB elimination in the United States. As indicated in the systematic review and meta-analysis by Alsdurf et al. (reference 2) this is not successfully happening within the current health care paradigm and therefore, task shifting into primary care is an important strategy. Regarding the use of qualitative methods, we have indicated the exploratory nature of our research topic in the introduction. At this point in time, it is widely recognized that qualitative methods are uniquely suited to exploring topics or applications for which little is known because data are collected in participants’ own words. We added a sentence at the end of the introduction. As mentioned in the introduction and methods, these results were used to design an educational program geared towards primary care providers (detailed methods can be found here: NCT04188041) 

For Explain how to ensure study rigor:

Your changes were not convincing to me. Criteria as well as action for each criterion are not mentioned

-We included additional citations in our document to demonstrate reasoning behind methodology and to demonstrate that methods used are established in qualitative research. Please refer to References #9, 13, and 15. Please see S1 file for our completed Consolidated Criteria for Reporting Qualitative Research document. We hope that these criteria are what the reviewer is referring to.

For Labeling the categories with a qualitative look:

Your changes were not convincing to me.

-As described, we used the process of immersion/crystallization (reference 14) to develop the code categories. This is an established qualitative data analysis technique. 

Needs editing, especially in the case of spacing.

-Thank you. This may be an issue when switching between word processing software. We have gone through and eliminated all of the additional spaces and hope that this issue is now resolved.

---

## [Editor Report · Decision Letter 3]

1 Apr 2022

Primary care providers’ and nurses’ knowledge, attitudes, and skills regarding latent TB infection testing and treatment: A qualitative study from Rhode Island

PONE-D-21-05814R3

Dear Dr. Szkwarko,

We’re pleased to inform you that your manuscript has been judged scientifically suitable for publication and will be formally accepted for publication once it meets all outstanding technical requirements.

Kind regards,

Hamidreza Karimi-Sari, MD

Academic Editor

PLOS ONE
---

## [Editor Report · Acceptance letter]

7 Apr 2022

PONE-D-21-05814R3 

Primary care providers’ and nurses’ knowledge, attitudes, and skills regarding latent TB infection testing and treatment: A qualitative study from Rhode Island 

Dear Dr. Szkwarko:

I'm pleased to inform you that your manuscript has been deemed suitable for publication in PLOS ONE. Congratulations! Your manuscript is now with our production department. 

Kind regards, 

on behalf of

Dr. Hamidreza Karimi-Sari 

Academic Editor

PLOS ONE